
# Dataset of airborne measurements of aerosol, cloud droplets and meteorology by tethered balloon during PaCE 2022

Viet Le[1], Konstantinos Doulgeris[1], Mika Komppula[1], John Backman[1], Gholamhossein Bagheri[2], Eberhard Bodenschatz[2], and David Brus[1]

[1]Finnish Meteorological Institute, Helsinki, FI-00560, Finland
[2]Max Planck Institute for Dynamics and Self-Organization, Am Fassberg 17, 37077 Göttingen, Germany

**Correspondence:** Viet Le (viet.le@fmi.fi)

**Abstract.**

Aerosol, cloud droplet, and meteorological measurements were carried out by the Finnish Meteorological Institute's payload onboard the tethered balloon systems during the Pallas Cloud Experiment 2022 in Finland. This dataset includes 21 flights between September 16[th] and October 10[th]. The observations include vertical profiles and time series of aerosol number concentration and size distribution; cloud droplet number concentration and size distribution; and meteorological parameters. This dataset has been uploaded to the common Zenodo PaCE 2022 community archive (https://zenodo.org/communities/pace2022/, last access: Jan 20, 2025). The dataset (Le et al., 2025) is available at: https://doi.org/10.5281/zenodo.14932882.

## 1 Background

The most pronounced impact of climate change is in the Arctic region, where the warming is nearly four times as large as the global average (Rantanen et al., 2022). This phenomenon is known as Arctic amplification and has been observed in paleoproxy reconstruction of past climate (Park et al., 2019), present-day instrumental observations (Bekryaev et al., 2010; Esau et al., 2023), and future predictions of climate models (Holland and Bitz, 2003; Davy and Outten, 2020). It is projected that Arctic sea ice during summers will completely melt by 2050 (Ono et al., 2022), leading to the disruption of Arctic ecosystems and inhabitants, as well as significant alterations to global climate patterns.

Early research has linked Arctic amplification to the sea ice albedo feedback (Manabe and Wetherald, 1975), which is a positive feedback that accelerates sea ice melting through the decrease of surface albedo due to sea ice loss. Additionally, this loss of sea ice destabilizes the boundary layer and enhances moisture availability and boundary layer convection (Kay and Gettelman, 2009; Vavrus et al., 2009; Philipp et al., 2020). As a result, there is an increase in Arctic low clouds in fall and winter as observed in recent decades (Cao et al., 2017; Philipp et al., 2020). Given the impact of clouds on the global radiation budget, many studies (Vavrus, 2004; Taylor et al., 2013) have confirmed the importance of cloud feedback in Arctic amplification. These abundant Arctic low clouds warm the surface by increasing the downward longwave radiation (Taylor et al., 2013; Cao et al., 2017), which in turn could promote more liquid cloud droplets (Tan and Storelvmo, 2019; Huang et al., 2021) that further enhance the downward longwave radiation.





Currently, the uncertainty of Arctic cloud feedback remains high in climate models (Forster et al., 2021). A major factor
contributing to this issue is the lack of reliable observations of aerosols and clouds. Aerosols can serve as cloud condensation
nuclei (Aitken, 1881), playing a crucial role in cloud formation and influencing cloud properties. Dada et al. (2022) and
Doulgeris et al. (2023) demonstrated that anthropogenic pollution has a significant impact on cloud properties, especially in
Arctic and sub-Arctic clean environments. The increase in anthropogenic pollution leads to a higher cloud condensation nuclei
concentration, which subsequently increases the cloud droplet number concentration ($N_d$), e.g. Twomey (1959). As a result,
the effective diameter (ED) of the cloud droplets decreases, altering the cloud's radiative properties by increasing its albedo.
Moreover, small cloud droplets also inhibit precipitation and prolong cloud lifetime, which ultimately enhances the cloud's
liquid water content (LWC). Despite the well-known impact of aerosols on cloud properties, quantifying the aerosol-cloud
interaction (ACI) remains highly uncertain, with only a slight reduction in uncertainty across the past six IPCC reports (Forster
et al., 2021). As a result, accurate observations of aerosols and clouds are essential for enhancing our understanding of this
interaction.

In the last three decades, airborne missions utilizing large crewed aircraft have yielded valuable information on aerosols and
their impact on clouds' microphysics over the Arctic (Curry et al., 1988; Borys, 1989; Browell et al., 1992; Lathem et al., 2013;
Ancellet et al., 2014; Abbatt et al., 2019). However, they are logistically and financially demanding, making them only feasible
for large-scale campaigns. Moreover, they are often not able to fly within hundreds of meters above ground, preventing them
from measuring the often low-level clouds in the Arctic. Tethered balloon system (TBS) is gaining recognition as a valuable
platform for collecting vertical distribution of aerosols and clouds in recent years (Hara et al., 2013; Ferrero et al., 2019;
Creamean et al., 2021; Pohorsky et al., 2024). It offers distinct advantages and limitations related to flight ceiling, profiling,
cost, and payload capacity. However, a key benefit of TBS is its ability to profile and hover at specific altitudes, with flight
durations of several hours possible, depending on the power available for instrumentation.

In this paper, we present our collected aerosol, cloud and meteorological measurements onboard TBS during the Pallas
Cloud Experiments (PaCE) campaign in 2022. The campaign was conducted in Pallas, northern Finland, from September 16[th]
to October 10[th], 2022, and is part of the ongoing PaCE campaigns that have been running for two decades (Brus et al., 2025).
In the following sections, we provide an overview of the campaign, instruments, and dataset measured by the FMI's payload.
These observations are crucial for identifying key processes related to aerosol and cloud interactions. Additionally, they provide
an important observational basis for models, ground-based remote sensing, and satellite data validation.

## 2  Campaign overview

Over the last two decades, the Finnish Meteorological Institute (FMI) has carried out PaCE campaigns in the subarctic region
of Finnish Lapland. These campaigns played a key role in providing extensive observations for clouds and aerosols research in
the Arctic (Komppula et al., 2005; Lihavainen et al., 2010; Doulgeris et al., 2020, 2022). In 2022, PaCE was held once again
in Pallas, Finland (Figure 1), with participation from several institutes across Europe, each deploying different instrumentation
and measurement platforms. The campaign utilized a variety of methods to gather comprehensive datasets on atmospheric





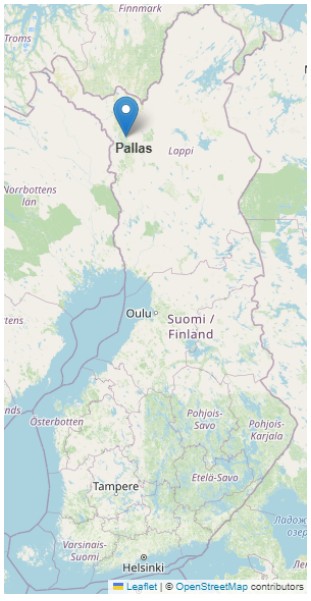

**Figure 1.** Location of PaCE 2022 campaign in Pallas, Lapland, northern Finland

properties, such as airborne in-situ measurements onboard uncrewed aerial systems (UAS) and TBS, as well as ground-based in-situ and remote sensing observations. More detailed descriptions of the campaign can be found in our overview PaCE 2022 paper by (Brus et al., 2025).

The data presented in this paper was measured by a FMI's payload onboard TBSs during the intensive part of the campaign, from September 16th to October 10th, 2022. During the first week from September 16th to September 23rd, the FMI's payload was onboard the Max Planck CloudKite (MPCK) platform (see Fig. 2) by the Max Planck Institute for Dynamics and Self-Organization (MPIDS) (Chavez-Medina et al., 2025; Schlenczek et al., 2025). The MPCK platform was flown during the PACE in a tandem configuration, i.e. a 250 m$^3$ helikite with a 34 m$^3$ helikite above it (both Desert Star Helikites Allsopp Helikites

Ltd.). These helikites, which are hybrids of helium balloons and kites, have stable flight behaviour in both calm and windy conditions with a minimum payload of up to 100 kg at 1 km above sea level. The flight altitude of the MPCK platforms can be controlled by reeling in and out the tether with a diesel winch. This unintentionally resulted in the measurements being affected by diesel emissions when the payload was near the ground.

    Starting from the second week of the measurement on September 30th, FMI's payload was onboard FMI's tethered balloon

system Aerostat by SkyDoc Systems Inc. (see Fig. 3). The Aerostat system consists of a tethered balloon, a winch, and a tether line (about 1 km in length), which is controlled by the winch. Unlike the MPCK platform, the Aerostat is operated by using an electric winch, ensuring no interference with the payload measurements. When inflated with 50 m$^3$ helium, the Aerostat provides a net lift capacity of approximately 37 kg under zero wind conditions at sea level. The FMI payload had a weight of about 15 kg and contained several instruments designed to measure meteorological parameters, as well as properties of cloud

and aerosol particles. A detailed description of these instruments will be provided in the next section.





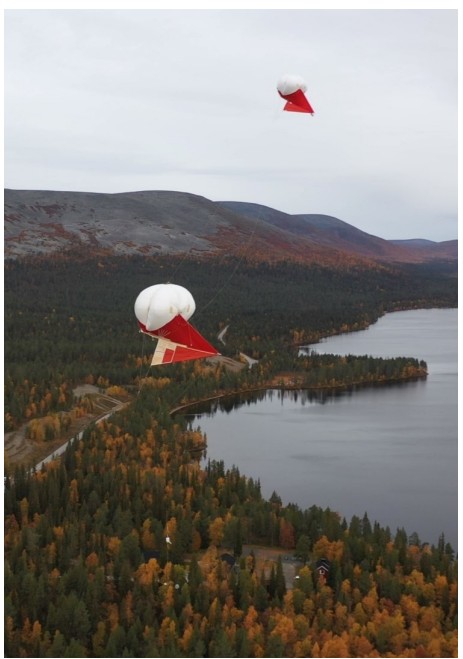

**Figure 2.** FMI's payload onboard the MPCK platform

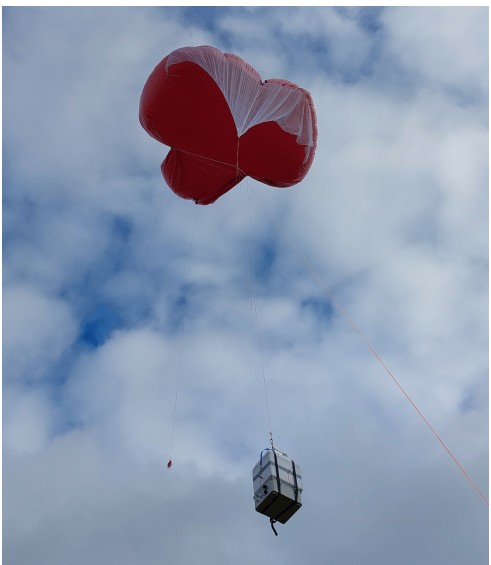

**Figure 3.** FMI's payload onboard onboard FMI's tethered balloon system Aerostat (SkyDoc Systems Inc.)

Both the MPCK platform and Aerostat were launched at the Pallasjärvi lake beach (68°01'23.2"N 24°09'48.8"E), with the launch site situated at an elevation of 276 meters above sea level. The location is relatively peaceful, about 500 m from the main



road. During the usual flights, the maximum distance of the MPCK platform and Aerostat from the launch site was around 700 m, and their maximum height above ground level was up to 1.5 km. During the measurement period, the weather was mild. The ambient temperature ranged from -1 to 6 °C, which minimized ice collection on the surfaces that could also lead to clogging of the instruments' inlets. The wind was moderate for most of the measurement period, averaging at 6.8 m s$^{-1}$. Several flying schemes have been utilized during the campaign, such as hovering at the altitude of the cloud layer or vertical profiling.

On October 10$^{th}$, an unexpected incident occurred during the TBS operation. A strong wind gust caused the tethered line of the balloon to be cut, resulting in the balloon and its payload flying away. The balloon ascended freely to an altitude of 8 km before bursting and gradually descending. It was subsequently recovered with the payload in full working condition, with all data successfully recorded. This event not only provided a unique dataset but also offered valuable lessons about flight safety for our future operations.




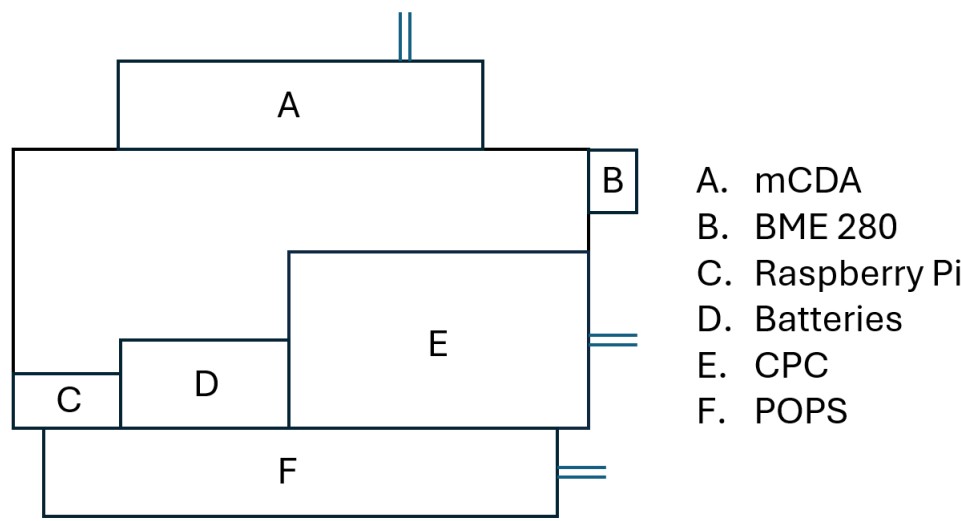

A. mCDA
B. BME 280
C. Raspberry Pi
D. Batteries
E. CPC
F. POPS

**Figure 4.** Schematic of the payload

## 3   Instrument overview

FMI built a custom battery-powered payload consisting of multiple instruments. The payload had dimensions of $32 \times 40 \times 50$
cm (length $\times$ width $\times$ height). It was attached to a 4 m line that was attached to the tethered balloon. The payload consisted
of a basic meteorological sensor (BME 280, Bosch Sensortec), a condensation particle counter (CPC, model 3007, TSI Inc.), a
portable optical particle spectrometer (POPS, Handix Inc.), and a mini cloud droplet analyzer (mCDA, Palas GmbH). All the
data were logged by a Raspberry Pi 3+ minicomputer using Python scripts to ensure that the time stamps were synchronized.
The BME, CPC, and POPS were logged at a rate of 1 Hz, while mCDA was logged at a rate of 10 Hz. The schematic of the
payload is shown in Fig. 4. The Raspberry Pi, CPC, and their batteries were placed inside a styrofoam box, while the remaining
instruments were housed in their own enclosures and attached to the styrofoam box on top and bottom.

The Bosch BME sensor was used to measure ambient conditions outside the payload, including pressure (p), temperature
(T), and relative humidity (RH). The BME sensor has a manufacturer-stated accuracy (at 25°C) of $\pm 1$ hPa for p, $\pm 0.5$°C for T,
and $\pm 3$ % for RH (Humidity sensor BME 280: https://www.bosch-sensortec.com/products/environmental-sensors/humidity-
sensors-bme280/, last access: 08 December 2024).

The CPC is a hand-held condensation particle counter that can measure the particle size range from 0.01 to about 1 μm. The
device was placed inside the styrofoam box, with a 10 cm length of conductive tubing serving as an inlet, extending through
the wall of the box to the outside. The CPC operates by drawing aerosol samples with laminar flow continuously through a
heated saturator and then a cooled condenser where the isopropyl alcohol vapors condense onto the aerosol particles. As those
particle droplets grow in size, they are counted by an optical detector. The isopropyl alcohol wick was resoaked before every
flight to ensure consistent measurements. The CPC calibration was done at FMI in the same way as described in Hämeri et al.



(2002); the uncertainty of $D_{50}$ value was $\pm 0.9$ nm for 10 nm particles. The total count was compared to a desktop, full-sized, more precise CPC (model 3772, TSI Corp.) with an accuracy of about 20 % when ambient air was sampled.

The POPS is an optical particle spectrometer for measurements of aerosol number concentrations and size distributions. The POPS uses a 405 nm diode laser to measure and size aerosol particles within the size range of 0.12–4.4 µm. Individual aerosol particles are pulled through an inlet nozzle, after which they intersect a 405 nm diode laser beam. The scattered light is focused onto a photomultiplier tube, which generates currents proportional to the scattered light, which is proportional to the diameter of the sampled particle. The POPS was enclosed in a plastic housing provided by the manufacturer and attached to the bottom of the styrofoam box. The POPS was calibrated by the manufacturer and used as such during the whole campaign.

The mCDA is a prototype mini cloud droplet analyzer by Palas GmbH. It can measure cloud droplets ranging from 1 to 100 µm in 256 bins up to 500 particles per cubic meter. The mCDA has a dimension of 160 x 110 x 290 mm, weighs 1.2 kg, and consumes less than 10 W of power. It was mounted on top of the styrofoam box with a 10 cm vertical inlet exposed to the environment, and the flow was maintained at 2.8 $\mathrm{Lmin^{-1}}$. The number size distribution of the sampled air was determined by the optical light scattering by individual particles. The mCDA was calibrated at FMI before and after the campaign using
MonoDust 1500 calibration standard and obeying the manufacturer's single size calibration procedure.



**Table 1.** Data description of csv files

| Column name | Description | Instrument |
|---|---|---|
| datetime (utc) | Date and time of the data point from all the instruments in UTC | Raspberry pi |
| temp_bme (C) | Ambient temperature ($^\circ$C) | BME280 |
| press_bme (hPa) | Ambient pressure (hPa) | |
| rh_bme (%) | Ambient relative humidity (%) | |
| N_conc_cpc (cm-3) | Particle number concentration ($cm^{-3}$) | CPC |
| press_cpc(hPa) | Inlet pressure (hPa) | |
| N_conc_pops (cm-3) | Particle number concentration ($cm^{-3}$) | POPS |
| press_pops (hPa) | Inlet pressure (hPa) | |
| flow_rate_pops (l/m) | Inlet flow rate ($Lmin^{-1}$) | |
| binX_pops (cm-3) | Aerosol number concentration ($cm^{-3}$) in binX, with X ranging from 1 to 16 | |
| binX_pops (dN/dlogDp) | Normalized aerosol concentration (dN/dlogDp) in binX, with X ranging from 1 to 16 | |
| binX_mcda (cm-3) | Number concentration ($cm^{-3}$) in binX,with X ranging from 1 to 175 | mCDA |
| pm1_mcda | Particulate matters with diameter less than 1 micron ($\mu g\,cm^{-3}$) | |
| pm25_mcda | Particulate matters with diameter less than 2.5 micron ($\mu g\,cm^{-3}$) | |
| pm10_mcda | Particulate matters with diameter less than 10 micron ($\mu g\,cm^{-3}$) | |
| binX_mcda (dN/dlogDp) | Normalized cloud droplet concentration (dN/dlogDp) in binX, with X ranging from 1 to 175 | |
| Nd_mcda (cm-3) | Total cloud droplet concentration ($cm^{-3}$) | |
| LWC_mcda (g/m3 ) | Liquid water content ($g\,m^{-3}$) | |
| MVD_mcda (um) | Median volume diameter (µm) | |
| ED_mcda (um) | Effective droplet diameter (µm) | |

## 4 Dataset overview, evaluation and quality control

The dataset contains measurements from the ground before being airborne until landing back on to the ground. Measurements below, in, and occasionally above clouds were recorded. For each flight, data from all the instruments was combined into a single file for release and further analysis. These files are given in ASCII comma-separated values (CSV) and NetCDF format.

Their names are FMI.TBS.b1.yyyyMMdd.hhmm.csv and FMI.TBS.b1.yyyyMMdd.hhmm.nc, where yyyyMMdd and hhmm indicate the date and time of the first measurement data point, respectively. The overview description of each file in csv format is shown in Table 1. All data was carefully monitored and quality controlled. Missing or bad values were set to -9999.9.

The normalized concentration (dN/dlogD$_p$) is calculated from POPS measurements in 16 PSL equivalent bins: 0.120 - 0.141 µm, 0.141 - 0.169 µm, 0.169 - 0.204 µm, 0.204 - 0.228 µm, 0.228 - 0.253 µm, 0.253 - 0.279 µm, 0.279 - 0.354 µm, 0.354 -

0.604 µm, 0.604 - 0.705 µm, 0.705 - 0.786 µm, 0.786 - 1.101 µm, 1.101 - 1.118 µm, 1.118 - 1.766 µm, 1.766 - 2.690 µm, 2.690 - 3.015 µm, 3.015 - 4.393 µm.

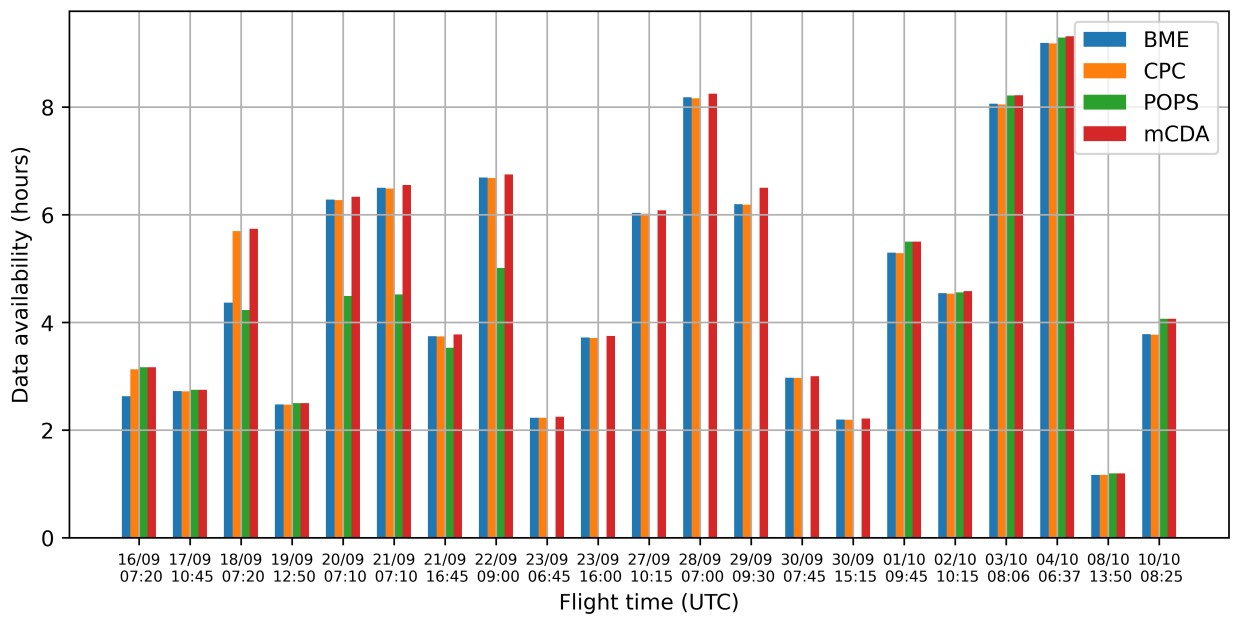

**Figure 5.** Data availability from all the instruments in the FMI's payload. The x-axis shows the flight starting time in UTC

From 16th of September to 2nd of October, the normalized concentration (dN/dlogD$_p$) was calculated from mCDA across 175 water-equivalent bins, with values ranging from 0.558 µm to 41.791 µm. On the 3rd of October, the mCDA was recalibrated, adjusting the 175 bins to span from 1.401 µm to 99.568 µm. The precise range of each size bin is provided in the data.

Additionally, total cloud droplet number concentration (N$_d$), liquid water content (LWC), median droplet volume diameter (MVD), and effective droplet volume diameter (ED) were also derived based on the measured cloud droplet concentration, following Doulgeris et al. (2020). It should be noted that these parameters were only available when the payload was in the cloud.

Figure 5 illustrates the measurement hours for each instrument across all 21 flights. On the 23rd of September, the internal

pump inside the POPS experienced a failure, resulting in erroneous data from the 23rd to the 30th of September, as mentioned previously. This issue was eventually identified and resolved, allowing POPS data to be available again starting from 1th October. Except for this period, data collected from all the instruments is available simultaneously during all flights. An example of these harmonized data is shown in Fig. 6. In general, the duration for each flight was determined based on the weather conditions, such as cloud cover and wind speed. This guarantees both the safety of the operation and the scientific

value of the data.

Figure 7 presents the meteorological conditions recorded by the payload across all the flights. The measured pressure ranged from 380 hPa to 1000 hPa, with the median pressure at around 900 hPa. The measured temperature varied from -22 °C to 23 °C with the median temperature at approximately 1 °C. The relative humidity spanned from 25 % to 100 %. Figure 8 illustrates LWC, MVD, ED, and normalized size distribution (dN/dlogD$_p$) derived from mCDA measurements across all the flights. A

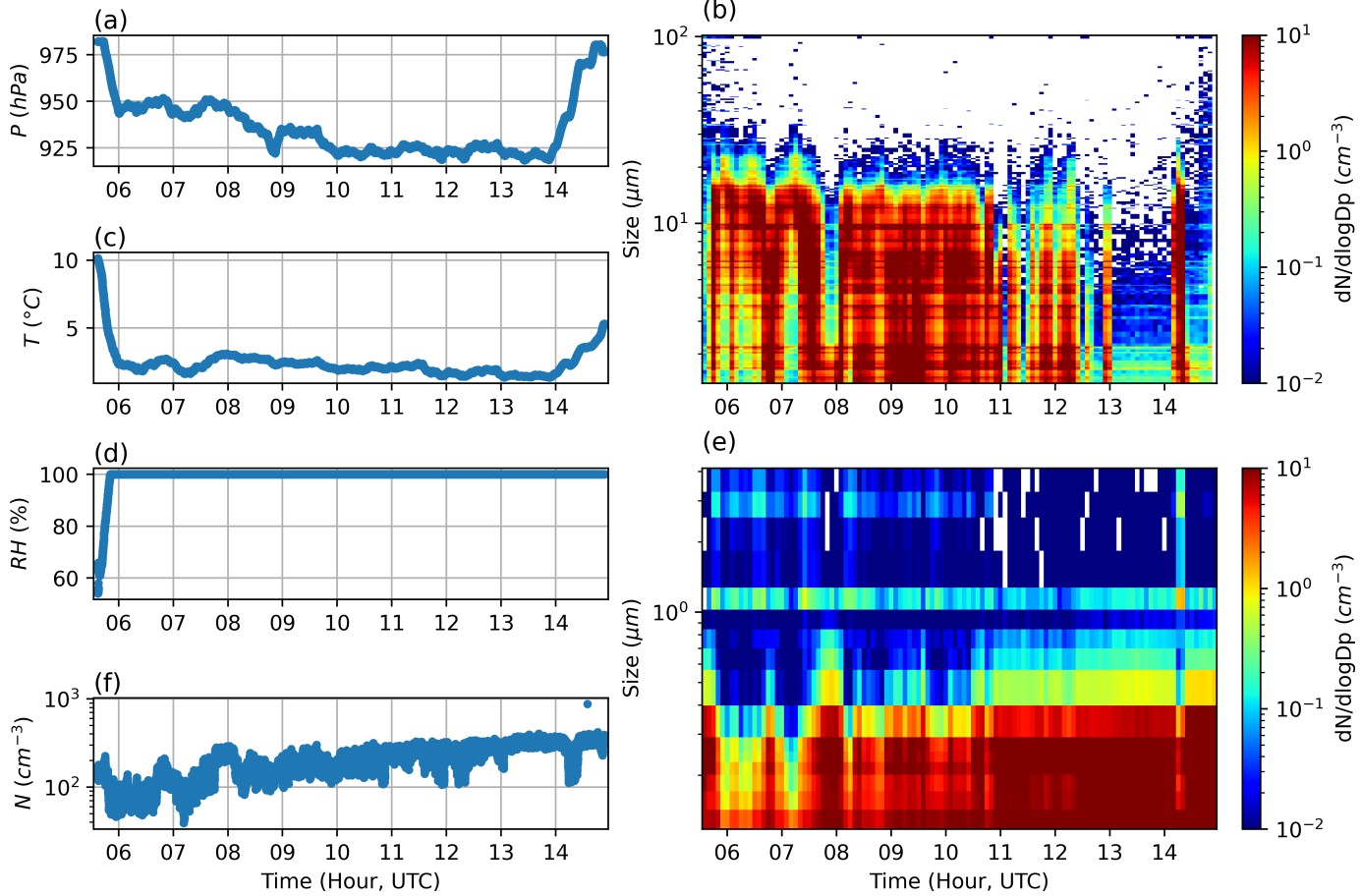

**Figure 6.** Measurements on the 4th of October flight from the FMI's payload. a) Pressure (hPa) measured by BME280, b) dN/dlogD$_p$ (cm$^{-3}$) calculated from mCDA measurements, c) Temperature (°C) measured by BME280, d) Relative humidity (RH) measured by BME280, e) dN/dlogD$_p$ (cm$^{-3}$) calculated from POPS measurements and f) Number concentration (cm$^{-3}$) from CPC

total of 19.34 hours of in-cloud measurements and 83.15 hours of no-cloud measurements were recorded. The averaged LWC was $3.4 \times 10^{-3} \pm 9 \times 10^{-3}$ g m$^{-3}$; MVD was 9.85 ± 3.96 μm, and ED was 8.63 ± 3.30 μm. Figure 9 shows the measured total number concentration from CPC, POPS and mCDA across all the flights. Each instrument has a different cut-off diameter range: 0.01 - 1 μm for CPC, 0.120 - 4.4 μm for POPS (PSL equivalent), and 0.558 - 41.791 μm and 1.401 - 99.568 μm for mCDA (water equivalent). In the whole campaign, when the payload was not in clouds, the averaged total number concentration

from CPC was 532.99 ± 954.63 cm$^{-3}$, and from POPS was 52.14 ± 84.72 cm$^{-3}$. Conversely, when the payload was in clouds, the averaged total number concentration from CPC was 241.15 ± 215.84 cm$^{-3}$, from POPS was 46.51 ± 42.50 cm$^{-3}$ and the averaged total N$_d$ from mCDA was 14.96 ± 20.22 cm$^{-3}$ .



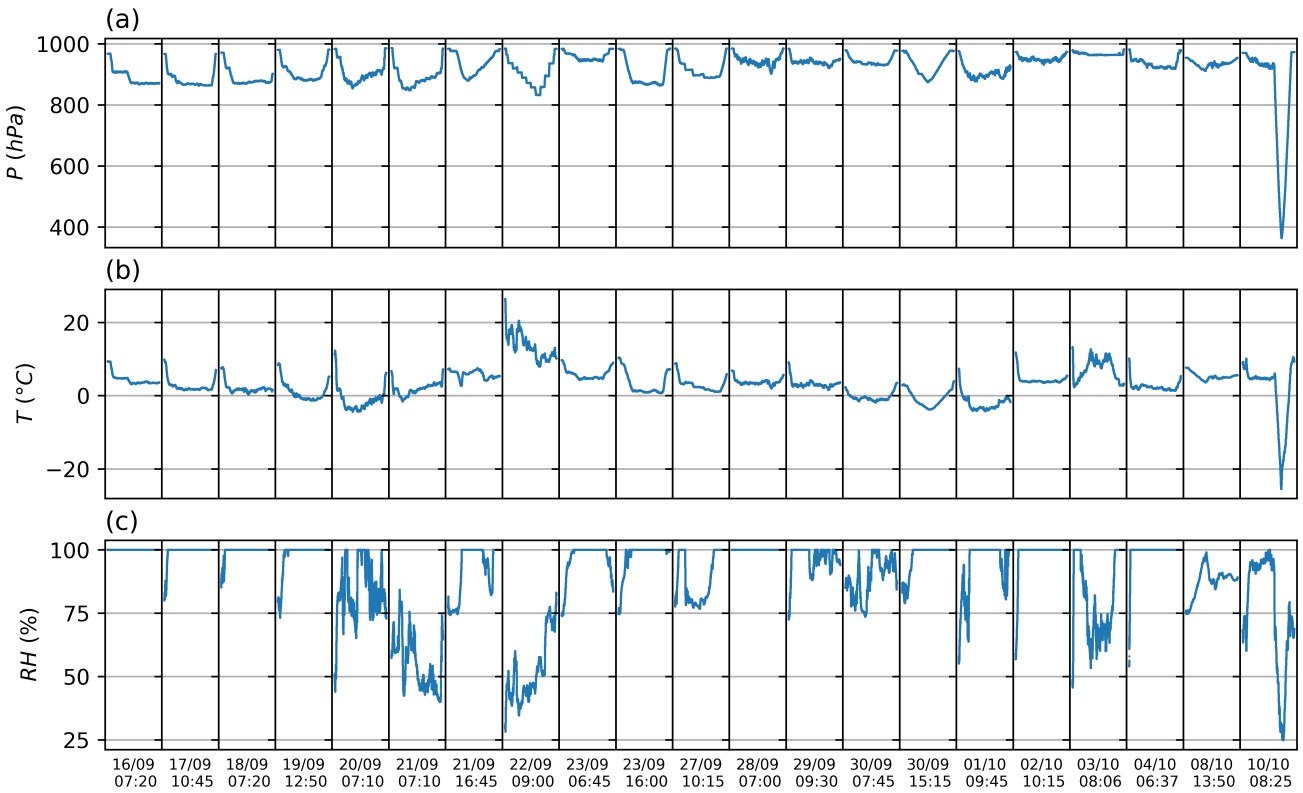

**Figure 7.** Overview of meteorological parameters measured by the FMI's payload: a) Pressure (hPa), b) Temperature (°C) and c) Relative humidity (RH) measured by BME280. The x-axis shows the flight starting time in UTC

**Figure 8.** Overview of a) Liquid water content (g m$^{-3}$), b) Median volume diameter (µm), c) Effective diameter (µm), and d) dN/dlogD$_p$ (cm$^{-3}$) calculated from mCDA measurements from all the flights. The x-axis shows the flight starting time in UTC

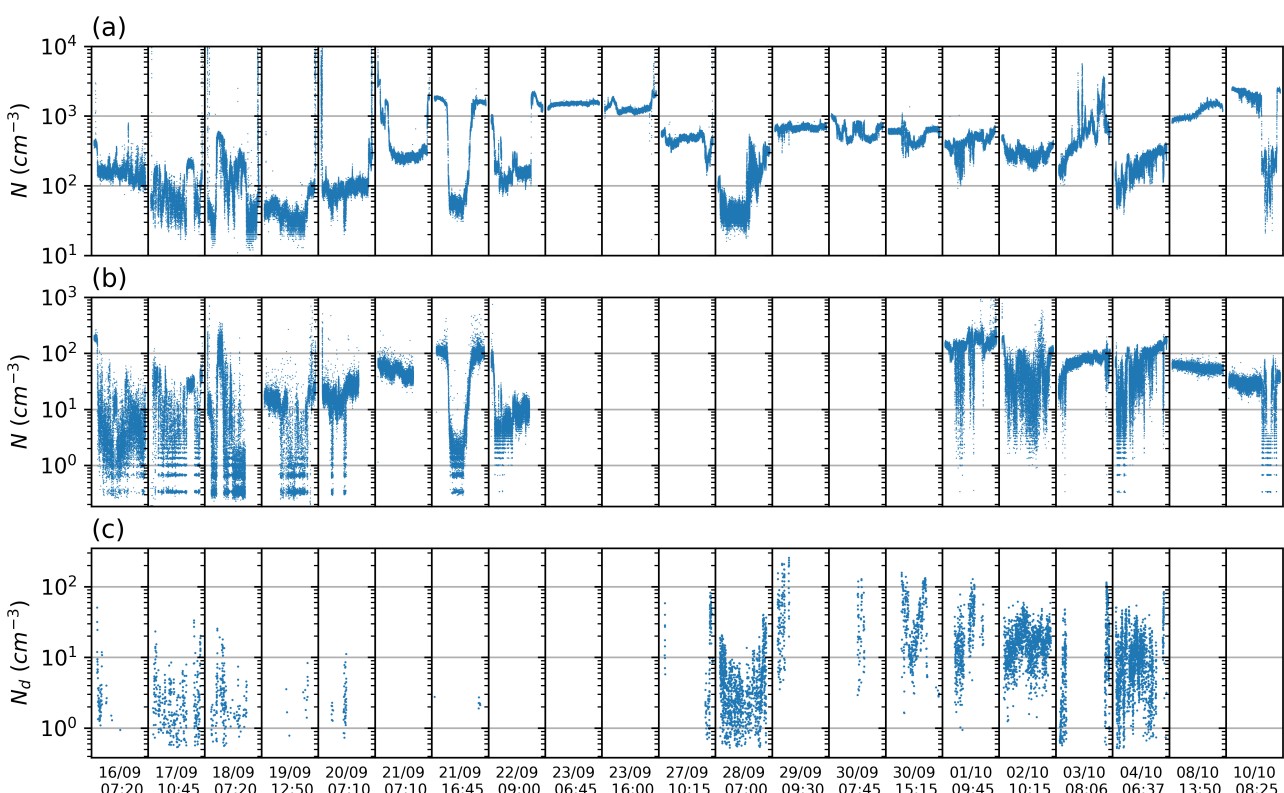

**Figure 9.** Overview of number concentration ($cm^{-3}$) from a) CPC b) POPS, and c) mCDA from all the flights. The x-axis shows the flight starting time in UTC



## 5 Summary

This paper provided an overview of the data obtained from the FMI's payload onboard TBSs during the PaCE 2022 campaign
from September to December 2022 in Pallas in the Finnish sub-Arctic. In Section 2, we provided an overview of the campaign
as well as the deployed tether balloon systems. In Section 3, we described the payload setup and instruments, including BME,
POPS, CPC and mCDA. Section 4 presented a description of the dataset, its processing procedure, and an overview of the
measured parameters. In summary, during the campaign, we measured the cloud properties in Pallas and found the averaged
$N_d$ of $14.96 \pm 20.22$ cm$^{-3}$ (with a median of 8.83 cm$^{-3}$), ED of $8.63 \pm 3.30$ μm (with a median of 8.37 μm) and LWC of
$3.4 \times 10^{-3} \pm 9 \times 10^{-3}$ g m$^{-3}$ (with a median of 0.95 g m$^{-3}$).

This dataset is part of the data collected during the PaCE 2022 campaign. In addition, other platforms such as unmanned
aerial vehicles, ground-based in-situ instruments, and ground-based remote sensing instruments were deployed concurrently.
Together, this diverse dataset provides a comprehensive set of observations of atmospheric properties across various research
topics. For instance, the cloud measurements from the TBSs can be used to validate cloud microphysical properties derived
from lidars and cloud radars, as outlined by (Frisch et al., 2002; Donovan et al., 2015; Vivekanandan et al., 2020).

*Data availability.* The collected dataset (Le et al., 2025) is available at https://doi.org/10.5281/zenodo.14932882. It was published at the Zenodo Open Science data archive, under a dedicated community Pallas Cloud Experiment – PaCE2022 (https://zenodo.org/communities/pace2022/, last access: 10 March 2025)

*Author contributions.* DB and KD planned and coordinated the FMI flights during PaCE 2022 campaign, all authors conducted the measure-
ments. VL and DB processed, analyzed, and quality-controlled FMI dataset. DB designed the payload. VL prepared the manuscript and all authors contributed to manuscript editing.

*Competing interests.* The authors declare that they have no conflict of interest.

*Acknowledgements.* The authors would like to express their gratitude to the Metsähallitus personnel, especially Mirka Hatanpää, for their invaluable support during the Pallas Cloud Experiment 2022.

*Financial support.* This work was supported by ACTRIS IMP GA 871115, ACTRIS-Finland funding through the Ministry of Transport and Communications, the Atmosphere and Climate Competence Center Flagship funding by the Research Council of Finland (Grants 337552). This project has also received funding from the European Union, H2020 research and innovation program (ACTRIS-IMP, the European Re-





search Infrastructure for the observation of Aerosol, Clouds, and Trace gases, Grant 871115). Financial support from the Magnus Ehrnrooth foundation is also greatly appreciated.





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
