# Peer review of "Dataset of airborne measurements of aerosol, cloud droplets and meteorology by tethered balloon during PaCE 2022"

_Earth System Science Data, 2025_

## Referee Comment (RC2)

Review of ESSD-2025-148: "Dataset of airborne measurements of aerosol, cloud droplets and meteorology by tethered balloon during PaCE 2022" by Viet Le, Konstantinos Doulgeris, Mika Komppula, John Backman, Gholamhossein Bagheri, Eberhard Bodenschatz, and David Brus

This is a short, but effective, description of measurements of aerosol, cloud droplet, and associated mean meteorology made with a balloon-borne system during the PaCE 2022 field campaign in northern Finland.

I don't have any major issues, but there are a number of minor points that would help clarity, and one or two typos etc.

Line 21-22: '…clouds warm the surface…which in turn could promote more liquid cloud droplets…' – it would be useful to include a brief explanation of how/why this might occur. More droplets (distinct from more liquid water) implies more CCN. Why does warming of the surface increase CCN concentrations?

Line 30: Point about the relationship between CCN concentration, droplet number concentration, effective diameter, and albedo is true, but…not necessarily the dominant factor for Arctic clouds, where changes to longwave radiative fluxes are often larger than for shortwave, depending on location and season. A good reference might be *Kay et al. 2016,: Recent Advances in Arctic Cloud and Climate Research. doi:10.1007/s40641-016-0051-9*.

Line 39: "…often not able to fly within hundreds of meters above ground." – true, but perhaps 'above the surface' would be more general, including ocean and sea ice.

Line 40: 'Tethered Balloon System (TBS) is…" -> "Tethered Balloon Systems (TBS) are…' – there's more than one system in use (even within this study).

Line45: 'onboard TBS' -> 'onboard a TBS'

Line 78: 'the maximum distance of' -> 'the maximum horizontal distance of' – just to avoid confusion with vertical distance.

Figure 4 – the block diagram of the layout of sensors on the instrument package is fine, but the addition of a photograph would help the reader visualize what it really looks like.

Line 90: 'It was attached to a 4m line that was attached to the tethered balloon' – maybe 'It was suspended below the balloon attached to a 4m line' or similar. Clarifies it is hung below the balloon, and avoids awkward repetition of 'attached to'.

Line 101: '…condensation particle counter that can measure the particle size range from 0.01 to about 1 µm.' – should make clear that the CPC counts the total number or particle within this size range, but does not measure their size.

Line 106: "as described in Hämeri et al. (2002)" – this paper is missing from the reference list.

Line 107: "the uncertainty of $D_{50}$ value was ±0.9 nm for 10 nm particles" – should provide a brief explanation about what $D_{50}$ is a measure of. Why is this relevant here? The particle size is much larger than the upper limit of aerosol size range, so presumably refers to droplet after working vapour has condensed on to it.

Line 110: "measure and size aerosol particles within the size range of 0.12–4.4 µm" – it would be useful to add information on the number of size channels here.

Line 140: "resulting in erroneous data from the 23rd to the 30th of September, as mentioned previously." – I can't see a previous mention of this.

Line 161: 'tether balloon' -> 'tethered balloon'